# Impact of the Gestational Diabetes Diagnostic Criteria during the Pandemic: An Observational Study

**DOI:** 10.3390/jcm10214904

**Published:** 2021-10-24

**Authors:** María Molina-Vega, Carolina Gutiérrez-Repiso, Fuensanta Lima-Rubio, María Suárez-Arana, Teresa María Linares-Pineda, Andrés Cobos Díaz, Francisco J. Tinahones, Sonsoles Morcillo, María J. Picón-César

**Affiliations:** 1Departmento de Endocrinología y Nutrición, Hospital Universitario Virgen de la Victoria, 29010 Málaga, Spain; molinavegamaria@gmail.com (M.M.-V.); mjpiconcesar@gmail.com (M.J.P.-C.); 2Laboratorio de Investigación Biomédica de Málaga, Hospital Universitario Virgen de la Victoria, 29010 Málaga, Spain; carogure@hotmail.com (C.G.-R.); santi.lima@hotmail.com (F.L.-R.); teresamaria712@gmail.com (T.M.L.-P.); 3Centro de Investigación Biomédica en Red (CIBER) de Fisiopatología de la Obesidad y Nutrición, Instituto Salud Carlos III, 28029 Madrid, Spain; 4Departmento de Obstetricia y Ginecología, Hospital Regional Universitario de Málaga, IBIMA, 29009 Málaga, Spain; dramariasuarez@gmail.com; 5Laboratorio de Análisis Clínico, Hospital Universitario Virgen de la Victoria, 29010 Málaga, Spain; andres.cobos.sspa@juntadeandalucia.es

**Keywords:** gestational diabetes, COVID-19, pregnancy

## Abstract

Objective: To analyze the effect of applying alternative diagnostic criteria for gestational diabetes mellitus (GDM) during the COVID-19 pandemic on GDM prevalence and obstetrical and perinatal outcomes, in comparison to usual diagnostic approaches. Methods: Data from women referred to GDM diagnosis from 1 September to 30 November 2019 were retrospectively collected (2019-group). The same data from the same period in 2020 were prospectively collected (2020-group). In both cases, a two-step diagnostic approach was used, the first step being a screening test (1 h 50 goral glucose tolerance test, OGTT). In 2019 it was followed by a 100 gr OGTT for diagnosis. In 2020, this was replaced by a blood test for the measurement of plasma glucose and HbA1c, according to alternative GDM diagnostic criteria during the COVID-19 pandemic. Results: From 237 women in the 2019 group, 40 (16.9%) were diagnosed with GDM, while from 255 women in the 2020 group, 37 (14.5%) had GDM (*p* = 0.470). More women in the 2020 group, in comparison to the 2019 group, were nulligravid (41.9% vs. 47.2%, *p* = 0.013), had a personal history of GDM (11.4% vs. 4.6%, *p* = 0.013) and had macrosomia in previous pregnancies (10.2% vs. 2.1%, *p* = 0.001). Obstetrical and perinatal outcomes were similar when comparing women with GDM to non-GDM women in the 2019 and 2020 groups and between GDM women and non-GDM women. Conclusion: In a Spanish population, GDM prevalence during the COVID-19 pandemic using the alternative diagnostic criteria was similar to that found in 2019 using the usual diagnostic criteria. Despite women referred for GDM diagnosis during the pandemic having more GDM risk factors, obstetrical and perinatal outcomes were comparable to those observed before the pandemic.

## 1. Introduction

The diagnosis of gestational diabetes is a source of controversy due to differences in the recommendations of scientific societies; general consensus about the best diagnosis strategy for gestational diabetes mellitus (GDM) has never been reached. The most controversial aspects are: (a) the need for universal versus selective screening (i.e., only those women with risk factors); (b) the two-step approach using a screening test followed by a diagnostic test versus the one-step approach using only a diagnostic OGTT; (c) the most appropriate test for diagnosis (100 versus 75 g OGTT); and even (d) the proposed cut-off points [1,2]. In the middle of this unsolved controversy, the COVID-19 pandemic has intruded.

Despite the likelihood of SARS-Cov-2 infection appearing to be no higher in pregnant women [3], the course of the disease has been shown to be worse in this population [4,5], and detrimental effects to the offspring have been described [6]. Therefore, considering pregnant women as a high-risk group has made it necessary to perform a temporary reformulation of GDM diagnosis, based on an isolated analytical determination of fasting plasma glucose (PG), random PG, and/or HbA1c [7,8,9,10,11,12,13], in order to avoid pregnant women spending long periods of time at the hospital.

For the moment, different authors have retrospectively analyzed the possible impact of applying these alternative criteria on GDM diagnosis using historical series, finding rates of undiagnosed cases as high as 30–50% [14,15], and that those who have maintained OGTT in some specific cases [9] achieve the least decrease in diagnoses [16]. However, to the best of our knowledge, nobody has published data about the prospective use of the alternative GDM diagnostic criteria during the COVID-19 pandemic.

The aim of this study is to evaluate the real-life impact of using proposed temporary alternative criteria during the COVID-19 pandemic in a population of pregnant women, and compare the results with the population attended to in the same period in 2019, in order to determine whether women with GDM diagnosed in each of these periods show differences regarding clinical characteristics and obstetrical and perinatal outcomes.

## 2. Materials and Methods

### 2.1. Subjects

From 1 September 2019 to 30 November 2019 and in the same period of 2020, 237 and 255 women, respectively, were referred from primary care centers to our Pregnancy and Diabetes Clinic (University Hospital Virgen de la Victoria, Málaga, Spain) to perform the diagnostic test for GDM. We chose to analyze data from the same period of the year, in order to avoid known seasonality and temperature influence over the GDM diagnosis in our population [17].

In 2019, GDM diagnosis was performed using a two-step strategy according to National Diabetes Data Group (NDDG) criteria [18]: firstly, a screening test (selective in women with risk factors during the first trimester of pregnancy, and universal in women between 24–26 weeks of pregnancy) with a 50 g oral glucose load (O’Sullivan test) was done in primary care centers. If post-load glucose was ≥140 mg/dL (7.8 mmol/L) in the screening test, a diagnostic 100 g oral glucose tolerance test (OGTT) was performed. The threshold for the diagnosis of GDM were 105 mg/dL (5.8 mmol/L) for fasting glucose and 190 mg/dL (10.6 mmol/L), 160 mg/dL (9.2 mmol/L) and 145 mg/dL (8.0 mmol/L) for 60, 120 and 180 min respectively and diagnosis of GDM was made when glucose met or exceeded these levels at two or more time points. This group will be called 2019-group.

In 2020, alternative criteria were applied based on recommendations of the Spanish Group of Diabetes and Pregnancy (GEDE) of the Spanish Diabetes Society (SED), and the Spanish Society of Gynecology and Obstetrics (SEGO), published in May 2020 [13]. They recommended maintaining the current diagnostic strategy: selective screening in the first trimester in women with risk factors, and universal screening between 24–28 weeks of gestation, and a two-step approach. However, if the situation of the pregnant woman or the conditions of the center did not allow it, both OGTTs could be replaced by an isolated blood test (either baseline or random) for the measurement of plasma glucose and HbA1c [13]. In our center, screening in primary care with the O’Sullivan test was maintained. In those with a pathological screening test, if fasting glycemia was ≥100 mg/dL (5.55 mmol/L) and/or HbA1c ≥ 5.9% (29.5 mmol/mol) during the first term of pregnancy or if fasting glycemia was ≥95 mg/dL (5.27 mmol/L) and/or HbA1c ≥ 5.7% (28.5 mmol/mol) during the second term of pregnancy, they were considered to have GDM. This group will be called the 2020-group. In is important to highlight that all patients referred to our clinic during this period for GDM diagnosis received dietetic and life habits advice (which is usually only given to women with GDM diagnosis).

Data about personal gynecological and obstetrical history, personal history of gestational diabetes or macrosomia, family history of diabetes, preconception weight and data about the delivery (week, instrumental, caesarean section), neonatal weight and length, macrosomia, and perinatal complications (obstetric trauma, hypoglycemia, jaundice requiring phototherapy, respiratory distress, hospitalization, perinatal death or NICU admissions) were recorded from digital clinical history. Weight and height were measured during the GDM diagnostic visit according to standardized procedures, and body mass index (BMI) was calculated as weight (kg)/height^2^ (m^2^).

### 2.2. Assay

Antecubital venous blood samples were collected after a 12-h fast at 8 a.m. in both the 2019 and 2020 groups. In the 2019 group, the women subsequently underwent a 100-g OGTT with a commercial preparation. During the process no physical exercise was permitted and women were not allowed to eat or drink except water. Post-load blood samples were collected at 60, 120 and 180 min. Vacuette^®^ FX Sodium Fluoride/Potassium Oxalate PREMIUM tubes (Greiner Bio One, Madrid, Spain) were used. Blood samples were centrifuged within 30 min of collection and analyzed with Dimension Vista Analyzer (Siemens AG, Berlin, Germany) using the glucose oxidase method (within-run and between-run precision was 1–2%). HbA1c was measured using HPLC (high pressure liquid chromatography) by the ADAMS A1c (HA-8180V) analyzer by Menarini in 2020 group.

### 2.3. Ethics

All patients participating in the present study and belonging to the 2019 cohort have signed an informed consent form and the work has the approval of the local ethics committee. In the case of the 2020 cohort, the follow-up in consultation was telematic and verbal consent was obtained for the use of their data for research purposes, as the patients were not allowed access to the hospital.

### 2.4. Statistical Analysis

Categorical variables were presented as frequencies and percentages, and continuous variables as mean ± standard deviation. Comparisons between the quantitative data were performed by Student’s *t*-test. Comparisons between the qualitative data were tested by Chi-Squared Test. A minimum sample size of 456 participants was required to detect a 7% difference in the birthweight percentile > 90% with a significance level of 5% and a power of 80%.

Perinatal complications, such as neonatal hypoglycemia, jaundice requiring phototherapy, hospitalization, respiratory distress, admission to the neonatal intensive care unit (NICU), perinatal death, and obstetric trauma were very rare. For this reason, a composite variable comprising any of the previously mentioned complications, called “any perinatal complications”, was created.

Results were considered significant if *p* < 0.050. Statistical analysis was done with SPSS (15.0 version for Windows; SPSS, Chicago, IL, USA).

## 3. Results

### 3.1. Characteristics of Study Population

Data were collected from 237 pregnant women in 2019 (2019-group) and from 255 patients in 2020, during the COVID-19 pandemic (2020-group). Gestational age at the time of the diagnostic test was slightly higher in 2019-group (25.51 ± 6.17 vs. 24.35 ± 6.85 weeks, *p* = 0.051), although the proportion of women in the first trimester of pregnancy (gestational age < 13 week) and in the second-third trimester of pregnancy was similar in both groups (first trimester: 6.8% vs. 9.8%, second-third trimester: 93.2% vs. 90.2%, *p* = 0.255).

Glucose levels after the 50 g glucose screening test performed at primary care centre were similar in both groups (160.72 ± 16.58 vs. 158.10 ± 24.38 mg/dL, *p* = 0.198) as well as fasting (PG) (84.68 ± 7.86 vs. 83.57 ± 11.70 mg/dL, *p* = 0.224). In the 2019-group, the plasma glucose level after OGTT was 160.30 ± 30.32 mg/dL at 60 min, 137.57 ± 29.75 mg/dL at 120 min and 109.08 ± 32.01 mg/dL at 180 min. In the 2020-group, the mean HbA1c level was 5.22 ± 0.34% (26.1 ± 1.7 mmol/mol).

The comparison between women in the 2019-group and 2020-group regarding GDM risk factors, obstetrical and perinatal outcomes is compiled in the Appendix A. In brief, no differences were found in age, in BMI before pregnancy, or in the presence of a family history of diabetes. A higher rate of nulligravid women was observed in 2019-group in comparison to 2020-group (47.2% vs. 41.9%, *p* = 0.013). In addition, a higher percentage of women was found to have had GDM in previous pregnancies (4.6% vs. 11.4%, *p* = 0.013) and to have given birth to newborns with macrosomia (2.1% vs. 10.2%, *p* = 0.001) in 2020-group in comparison to 2019-group. The number of risk factors for GDM present in the 2019-group was lower than in the 2020-group (1.0 ± 0.85 vs. 1.33 ± 1.00, *p* = 0.003). Obstetrical and perinatal outcomes, when comparing 2019-group to 2020-group were similar.

### 3.2. Prevalence of GDM in 2019-Group vs. 2020-Group

In the 2019-group, 40 women were diagnosed with GDM after glucose overload (16.9%), two of them in the first trimester of pregnancy, whereas in the 2020-group the diagnosis of GDM was made in 37 women (14.5%), four of them in the first trimester of pregnancy. Despite the prevalence of GDM being 2.4% lower in 2020 in comparison to 2019, this difference was not statistically significant in the total sample (*p* = 0.470), neither in the first trimester (*p* = 0.757) nor in the second-third trimester of gestation (*p* = 0.407).

### 3.3. Comparison between GDM vs. Non-GDM Women in 2019-Group and 2020-Group

In the 2019-group there was no significant difference in the classical risk factors for GDM when comparing GDM vs. non-GDM women, while in the 2020-group, patients diagnosed with GDM had higher BMI and a higher prevalence of personal history of macrosomia in previous pregnancies than non-GDM women (Table 1).

Obstetric outcomes of women with GDM were similar to those of women without GDM in 2019 group, except for weight gain. Women in the 2019-group who were diagnosed with GDM gained less weight than those without GDM (6.88 ± 5.23 vs. 10.61 ± 5.00, *p* < 0.001), while in 2020-group weight gain was similar in both groups (9.79 ± 7.09 vs. 10.39 ± 5.66, *p* = 0.672). However, women with GDM in 2020-group presented a higher BMI before pregnancy and at prepartum than those without GDM. Perinatal data and the rate of any perinatal complication were similar between GDM and non-GDM women in both 2019 and 2020 (Table 1).

### 3.4. Comparison between GDM in 2019-Group and 2020-Group

When GDM women from 2019-group and 2020-group were compared (Table 2), we found that women in 2020-group were less frequently nulligravid (50% vs. 21.6%, *p* = 0.006) and that they had a higher prevalence of macrosomia in previous pregnancies (2.5% vs. 29.7%, *p* = 0.004). Regarding obstetric and perinatal outcomes, we did not find significant differences between groups, with similar prevalence of any perinatal complications.

### 3.5. Comparison between Non-GDM in 2019-Group and 2020-Group

We analyzed the group of pregnant women with a negative diagnosis for GDM in both groups. Data are shown in Table 3. Women in 2020-group had a higher prevalence of GDM and macrosomia in previous pregnancies. No differences in obstetric or perinatal outcomes were found between women who tested negative on the 2019 OGTT and those who tested negative on the proposed alternative diagnostic method for the pandemic, with similar rates of any perinatal complications.

## 4. Discussion

In our population, when comparing data on the diagnosis of GDM and obstetric and perinatal outcomes in women referred for GDM diagnosis from September to November of 2019, using the usual GDM diagnosis, and in the same period of 2020, using alternative criteria during the COVID-19 pandemic, we found: firstly, a prevalence of GDM 2.4% lower in 2020 than in 2019, but not statistically significant; secondly, that women in 2020 had more GDM risk factors than those in 2019 in the whole population, in GDM women, and in non-GDM women; and finally, no differences in obstetric and perinatal outcomes were found between the 2019-group and 2020-group, either when comparing those diagnosed with GDM and those or when comparing women without GDM and women with GDM in the two groups.

Since the COVID-19 pandemic began, concern about maintaining social distance and avoiding infection have influenced GDM diagnosis. Despite consensus regarding the convenience of reducing OGTT performance during the COVID-19 pandemic, no agreement has been reached regarding an alternative diagnostic criterion for GDM. For example, and taking into account only criteria for women in their second term of pregnancy: the Royal College of Obstetricians and Gynaecologists (RCOG) recommended using fasting PG ≥ 100 mg/dL (5.6 mmol/L), HbA1c ≥ 5.7% (28.5 mmol/mol) and/or random PG ≥ 162 mg/dL (9 mmol/L) to diagnose GDM [7]; In Canada [8], those pregnant women with random PG ≥ 200 mg/dL (11 mmol/L) and/or HbA1c ≥ 5.7% (28.5 mmol/mol) have been considered to have GDM; the Australian Diabetes in Pregnancy Society has established that those women with fasting PG ≥ 92 mg/dL (5.1 mmol/L) or with personal history of GDM should be diagnosed with GDM. In women with fasting PG 84–91 mg/dL (4.7–5.0 mmol/L), OGTT has to be performed, and a fasting PG ≤ 83 mg/dL (4.6 mmol/L) excludes GDM [9]. In Japan [11] and France [12] similar criteria for GDM diagnosis have been adopted: fasting PG ≥ 92 mg/dL (5.1 mmol/L) and/or HbA1c ≥ 5.7% (28.5 mmol/mol) (in addition, in Japan, random PG ≥ 196 mg/dL (10.8 mmol/L). The European Society of Endocrinology [10] recommend considering those women with random PG ≥ 162 mg/dL (9 mmol/L), fasting PG ≥ 100 mg/dL (5.6 mmol/L) and/or HbA1c ≥ 5.7% (28.5 mmol/mol) positive for GDM; finally, criteria established by Spanish societies have been provided in the material and methods section.

Many authors have retrospectively applied the alternative criteria proposed during the COVID-19 pandemic to historical cohorts to evaluate their effect on GDM diagnosis. Most of them have found a significant reduction of GDM prevalence. McIntyre et al. [16] compared the effect of using criteria for GDM recommended by the United Kingdom (UK), Canada and Australia for use during the COVID-19 pandemic in a cohort of 5974 HAPO study women, finding that in the UK prevalence decreased from 12.9% to 2.5%, in Canada from 9.3% to 1.7%; in Australia, it only decreased 25%, from 17% to 12.7%. Van Gemert et al. [15] examined the percentage of women that have not been diagnosed using Australian temporary criteria during the COVID-19 pandemic in a historical cohort of 16,522 women with a prevalence of GDM of 12.2%. They found that, considering women with fasting glucose ≤ 83 mg/dL (4.6 mmol/L) as not having GDM, 29% would have been underdiagnosed. Issa et al. [19] did not obtain positive results either. They applied RCOG criteria in 205 women with GDM attending their clinic in 2019, finding 53.5% of them missed GDM diagnosis. Similarly, Kasuga et al. [11] found that after applying the COVID-19 pandemic criteria in a cohort of 264 Japanese women diagnosed with GDM according to IADPSG criteria, 160 (60.6%) of them would be classified as not having GDM. van-de-l’Isle et al. [20], have done a different analysis, but with similar results. They compared a retrospective cohort of pregnant women screened for GDM using NICE criteria with a prospective cohort using RCOG recommendations during the COVID-19 pandemic. The rate of women identified as having GDM significantly decreased, from 7.7% using NICE criteria to 4.2% using RCOG criteria. Some of the patients that were considered as not having GDM using RCOG criteria were later retested using NICE criteria, finding 20.4% to have GDM.

As previously described, we hoped to find a lower prevalence of GDM in 2020-group. However, despite the prevalence of GDM in 2020 being lower, the differences found were not statistically significant (16.9 vs. 14.5% respectively; *p* = 0.470). This may be due to the maintenance of OGTT, as concluded by McIntyre et al. [16] in reference to the less detrimental effect of Australian criteria on GDM prevalence.

It is important to remark that GDM prevalence was similar between the groups, despite the women from the 2020-group having more risk factors for GDM. It is possible that lower sensitivity of the diagnostic method used could be the cause, and that the actual prevalence of GDM in the 2020-group should have been higher than found by us. However, as women in 2020 were more frequently multiparous than those in 2019, it is reasonable for them to have a higher prevalence of macrosomia or GDM in previous pregnancies. The usefulness of different approaches for GDM diagnosis has been reported with contradictory results. d’Emden et al. [14] found fasting PG > 83 mg/dL (4.6 mmol/L) to have 77% specificity and 54% sensitivity for GDM diagnosis. For their part, Nachtergaele et al. [21] concluded that, if pregnant women with a history of hyperglycemia in pregnancy are considered to have GDM and, in the rest of them, OGTT is only performed in those with fasting PG 84–91 mg/dL (4.7–5.0 mmol/L), it is possible to avoid the performance of more than 80% of OGTTs, and identify with a sensitivity of 72% and only 10.2% false negatives the women at the highest risk of adverse pregnancy outcomes. However, other authors do not agree about the convenience of alternative diagnostic criteria for GDM [22,23].

On the other hand, we found a lower weight gain during pregnancy in women with GDM in 2019 vs. non-GDM women, while in 2020 weigh gain was similar between groups. Consequently, weight gain was higher, but not statistically significant, in GDM-women in 2020-group vs. 2019-group (9.8 ± 7.1 vs. 6.9 ± 5.2; *p* = 0.089). In addition, GDM women from 2020-group had a higher BMI, though not to a statistically significant degree, before pregnancy than GDM women from the 2019-group (30.7 ± 5.9 vs. 28.9 ± 5.9 kg/m^2^, *p* = 0.169). This could be due to the observed weight gain during lockdown in the Spanish population [24], especially in women and in those people previously overweight and obese.

Despite possible changes in the prevalence of GDM, the most important parameter to take into account, when evaluating the convenience of new diagnostic criteria for GDM is whether we are able to identify those women who are going to have more adverse obstetrical and perinatal outcomes. Regarding how the lower detection of GDM and the high rate of false negatives with the COVID-19 DMG diagnostic criteria could affect obstetric and perinatal outcomes, authors have reported different results.

McIntyre et al. [16] found that women who were false negative according to UK and Canada criteria had worse obstetric and perinatal outcomes, while those false negative according to Australian criteria had similar obstetric and perinatal outcomes to true negatives. They attribute these differences to the maintenance of OGTT in selected cases in the Australian criteria, while the UK and Canada criteria exclude OGTT performance. Nachtergaele et al. [25] retrospectively applied French-speaking Society of Diabetes criteria for GDM diagnose during the COVID-19 pandemic in a population of 7334 women. They found that patients classified as false negative had lower glucose levels in OGTT, HbA1c and pre-pregnancy BMI. Despite 40% of false negatives needing insulin, obstetric and perinatal outcomes were similar between true positives and false negatives. However, it is important to highlight that when retrospective data are analyzed, false-negative patients were actually treated as GDM patients, so obstetric and perinatal outcomes are not the same as if they had not been treated. Although they did not evaluate obstetric and perinatal outcomes, Kasuga et al. [11], similarly to Nachtergale et al. [25], found false negative patients to have a lower pre-pregnancy BMI than true positive. We also found women non-GDM to have a lower pre-pregnancy BMI than GDM women in the 2020-group. On the contrary, they reported that 1 h and 2 h post-load glucose levels were higher in false negative patients than in true positive, which could suggest that those false negative women could have a high risk of hyperglycemia. However, no differences in insulin requirements were observed between groups. In our population, we also found no differences in the percentage of GDM women requiring insulin therapy in either the 2019-group or in the 2020-group.

We did not observe differences regarding obstetrical and perinatal outcomes when comparing GDM women from 2019-group vs. 2020-group. On the other side, as with McIntyre et al. [16] using Australian criteria, and Nachtergale et al. [25], we did not observe a higher rate of obstetric and perinatal adverse outcomes in those women classified as not having GDM in the 2020-group in comparison to non-GDM women from the 2019-group. These results are perhaps the most relevant, because they show that those patients who were considered negative and therefore did not receive treatment even though they presumably needed it did not have any more adverse obstetrical and perinatal outcomes. Because of the concern that there might be women with GDM who would remain undiagnosed, and based on recommendations of GEDE, SED and SEGO [13], we decided to implement dietary advice for all patients who had a positive screening test regardless of the result of their diagnostic test. It is possible that this advice could have prevented some adverse events in these women, since, after all, they received dietary treatment even though their blood glucose levels were not monitored. In addition, unlike other authors, we kept the two-step diagnostic strategy, because the first step was performed in the primary care center, where the volume of patients was lower and the safety of the women could be guaranteed. Thus, all women were at high pretest probability of GDM. We think that these complementary safety tools have been key to minimizing the effect of possible underdiagnosis.

Among the strengths of our work, we can highlight that it is the first study to date to compare perinatal and prospective data on the application of the alternative criteria proposed for the pandemic. As weaknesses, we can mention that we do not have a control group, since the same criteria were applied to all patients. Ideally, patients should have been randomised to be diagnosed with one of the two methods; however, it did not seem ethical to offer women a diagnostic method that could potentially have led to under-diagnosis. Although an attempt was made to mitigate this by comparison with a historical group of similar dates to avoid environmental influences on the outcome, we have no medium to long term data on the children of false negative mothers.

## 5. Conclusions

In conclusion, when comparing the use of the usual criteria before the pandemic to the use of the alternative diagnostic criteria for GDM during the COVID-19 pandemic, our population did not see a significant reduction in GDM prevalence. Women diagnosed with GDM and women not diagnosed with GDM in 2019 and in 2020 were similar in regard to obstetric and perinatal outcomes, with those women referred for GDM diagnosis in 2020 having more GDM risk factors. However, we should be cautious about generalising the results obtained to other populations. More prospective studies analyzing the impact of using different alternative diagnostic criteria in different populations and including evaluation of obstetrical and perinatal outcomes are needed in order to identify the best options for possible future emergency situations.

## Figures and Tables

**Table 1 jcm-10-04904-t001:** Comparison between GDM vs. non-GDM women in 2019-group and 2020-group: GDM risk factors, obstetrical and perinatal outcomes.

	2019-Group			2020-Group		
	GDM (40)	Non-GDM (197)	*p*	GDM (37)	Non-GDM (218)	*p*
**GDM risk factors**						
Age (years)	33.3 ± 5.6	33.2 ± 5.1	0.946	35.9 ± 4.8	33.6 ± 5.1	0.118
BMI before pregnancy (kg/m^2^)	28.9 ± 5.9	27.5 ± 6.9	0.268	30.7 ± 5.9	26.6 ± 6.0	<0.001
Nulligravid women	20 (50%)	102 (51.8%)	0.931	8 (21.6%)	99 (45.4%)	0.007
Family history of diabetes	14 (35%)	45 (22.8%)	0.185	13 (35.1%)	66 (30.3%)	0.591
GDM in a prior pregnancy	4 (10%)	7 (3.6%)	0.124	8 (21.6%)	21 (9.6%)	0.166
Previous macrosomia	1 (2.5%)	4 (2%)	1.000	11 (29.7%)	15 (6.9%)	0.002
**Obstetrical data**						
Maternal weight gain (Kg)	6.9 ± 5.2	10.6 ± 5.0	<0.001	9.8 ± 7.1	10.4 ± 5.6	0.672
BMI prepartum (kg/m^2^)	31.5 ± 5.5	31.3 ± 6.1	0.886	33.6 ± 5.7	30.1 ± 5.5	0.008
Hypertensive disorders of pregnancy ^a^	1 (2.5%)	12 (6.1%)	0.701	5 (13.5%)	13 (6%)	0.064
Gestational week at birth	39.1 ± 1.4	38.8 ± 1.9	0.438	37.6 ± 4.3	38.9 ± 1.7	0.096
Type of delivery			0.842			0.478
Non-instrumental Instrumental Cesarean	20 (50%)	103 (52.3%)		15 (40.5%)	102 (46.8%)	
	5 (12.5%)	19 (9.6%)		4 (10.8%)	31 (14.2%)	
	10 (25%)	53 (26.9%)		11 (29.7%)	47 (21.6%)	
**Perinatal data**						
Pretermbirth ^b^	1 (2.5%)	14 (7.1%)	0.678	3 (8.1%)	16 (7.3%)	0.859
Birthweight (g)	3249 ± 445	3273 ± 572	0.812	3338 ± 628	3332 ± 554	0.956
Birthlength (cm)	50.1 ± 1.9	49.9 ± 2.3	0.583	50.3 ± 2.6	50.5 ± 2.6	0.709
Birthweight pc	52.5 ± 28.1	55.3 ± 30.4	0.615	60.4 ± 29.8	57.2 ± 30.3	0.605
Birthweight > 90th pc	4 (10%)	29 (14.7%)	0.374	6 (16.2%)	34 (16.6%)	0.703
Birthweight < 3rd pc	0	6 (3%)		0	4 (1.8%)	
**Perinatal complications**						
Any perinatal complication	6 (15%)	22 (11.2%)	0.494	5 (13.5%)	18 (8.3%)	0.302
• Neonatal hypoglycemia	2 (5%)	2 (1%)	0.074	1 (2.7%)	4 (1.8%)	0.725
• Jaundice requiring phototherapy	3 (7.5%)	9 (4.6%)	0.441	3 (8.1%)	8 (3.7%)	0.219
• Hospitalization	4 (10%)	12 (6.1%)	0.369	4 (10.8%)	7 (3.2%)	0.035
• Respiratory distress syndrome	0	4 (2%)	0.363	3 (8.1%)	5 (2.3%)	0.061
• NICU admission	0	7 (3.6%)	0.226	2 (5.4%)	2 (0.9%)	0.042
• Perinatal death	0	2 (1%)	0.522	0	2 (0.9%)	0.559
• Obstetric trauma	1 (2.5%)	1 (0.5%)	0.209	0	1 (0.5%)	0.680

GDM: gestational diabetes mellitus; pc: percentile; NICU: Neonatal Intensive Care Unit ^a^: Includes: pre-pregnancy hypertension, gestational hypertension, preeclampsia ^b^: Gestational age at birth ≤ 36 weeks. *p*-values refer to the comparison between women with and without diabetes in the same year.

**Table 2 jcm-10-04904-t002:** Comparison between GDM women in 2019-group vs. 2020-group: GDM risk factors, obstetrical and perinatal outcomes.

GDM Women			
	2019-Group	2020-Group	*p*
**GDM risk factors**			
Age (years)	33.3 ± 5.7	35.0 ± 4.9	0.157
BMI before pregnancy (kg/m^2^)	28.9 ± 5.9	30.7 ± 5.9	0.195
Nulligravid women	20 (50%)	8 (21.6%)	0.006
Family history of diabetes	14 (35%)	13 (35.1%)	0.878
GDM in a prior pregnancy	4 (10.0%)	8 (21.6%)	0.313
Previous macrosomia	1 (2.5%)	11 (29.7%)	0.004
**Obstetrical data**			
Maternal weight gain (Kg)	6.9 ± 5.2	9.8 ± 7.1	0.089
Insulin requirement (%)	27.5	27.0	0.963
BMI prepartum (kg/m^2^)	31.5 ± 5.5	33.6 ± 5.7	0.169
Hypertensive disorders of pregnancy ^a^	1 (2.5%)	5 (13.5%)	0.052
Gestational week at birth	39.1 ± 1.4	37.6 ± 4.3	0.050
Type of delivery			0.782
Non-instrumental Instrumental Cesarean	20 (50%)	15 (40.5%)	
	5 (12.5%)	4 (10.8%)	
	10 (25%)	11 (29.7%)	
**Perinatal data**			
Pretermbirth ^b^	1 (2.5%)	2 (5.4%)	0.474
Birthweight (g)	3249 ± 445	3338 ± 628	0.507
Birthlength (cm)	50.1 ± 1.94	50.30 ± 2.6	0.784
Birthweight pc	52.5 ± 28.1	60.4 ± 29.8	0.282
Birthweight > 90th pc	4 (10%)	6 (16.2%)	0.310
Birthweight < 3rd pc	0	0	
**Perinatal complications**			
Any perinatal complication	6 (15%)	5 (13.5%)	0.852
• Neonatal hypoglycemia	2 (5%)	1 (2.7%)	0.603
• Jaundice requiring phototherapy	3 (7.5%)	3 (8.1%)	0.921
• Hospitalization	4 (10%)	4 (10.8%)	0.907
• Respiratory distress syndrome	0	3 (8.1%)	0.066
• NICU admission	0	2 (5.4%)	0.136
• Perinatal death	0	0	-
• Obstetric trauma	1 (2.5%)	0	0.333

GDM: gestational diabetes mellitus; pc: percentile; NICU: Neonatal Intensive Care Unit ^a^: Includes: pre-pregnancy hypertension, gestational hypertension, preeclampsia, ^b^: Gestational age at birth ≤ 36 weeks.

**Table 3 jcm-10-04904-t003:** Comparison between non-GDM women in 2019-group vs. 2020-group: GDM risk factors, obstetrical and perinatal outcomes.

Non GDM Women			
	2019-Group	2020-Group	*p*
**GDM riskfactors**			
Age (years)	33.2 ± 5.1	33.6 ± 5.1	0.446
BMI before pregnancy (kg/m^2^)	27.5 ± 6.7	26.6 ± 6.0	0.140
Nulligravid women	102 (51.7%)	99 (45.4%)	0.107
Family history of diabetes	45 (22.8%)	66 (30.3%)	0.327
GDM in a prior pregnancy	7 (3.5%)	21 (9.6%)	0.020
Previous macrosomia	4 (2%)	15 (6.8%)	0.034
**Obstetrical data**			
Maternal weight gain (kg)	10.61 ± 5.00	10.38 ± 5.66	0.727
BMI prepartum (kg/m^2^)	31.32 ± 6.10	30.17 ± 5.56	0.112
Hypertensive disorders of pregnancy ^a^	12 (6.1%)	13 (6.0%)	0.970
Gestationalweek at birth	38.85 ± 1.85	38.95 ± 1.72	0.609
Type of delivery			0.204
Non-instrumental Instrumental Cesarean	103 (52.3%)	102 (46.8%)	
	19 (9.6%)	31 (14.2%)	
	53 (26.9%)	47 (21.6%)	
**Perinatal data**			
Pretermbirth ^b^	14 (7.1%)	16 (7.4%)	0.058
Birthweight (gr)	3273 ± 572	3332 ± 554	0.332
Birthlength (cm)	49.90 ± 2.37	50.52 ± 2.64	0.307
Birthweight pc	55.34 ± 30.38	57.26 ± 30.37	0.561
Birthweight > 90th pc	29 (14.7%)	34 (15.6%)	0.636
Birthweight < 3rd pc	6 (3%)	4 (1.8%)	
**Perinatal complications**			
Any perinatal complication	22 (11.1%)	18 (8.3%)	0.316
• Neonatal hypoglycemia	2 (1%)	4 (1.8%)	0.485
• Jaundice requiring phototherapy	9 (4.6%)	8 (4%)	0.645
• Hospitalization	12 (6.1%)	7 (3.2%)	0.161
• Respiratory distress syndrome	4 (2%)	5 (2.3%)	0.854
• NICU ^C^admission	7 (3.6%)	2 (0.9%)	0.066
• Perinatal death	2 (1%)	2 (0.9%)	0.919
• Obstetric trauma	1 (0.5%)	1 (0.5%)	0.943

GDM: gestational diabetes mellitus; pc: percentile; NICU: Neonatal Intensive Care Unit ^a^: Includes: pre-pregnancy hypertension, gestational hypertension, preeclampsia ^b^: Gestational age at birth ≤ 36 weeks.

## Data Availability

The data that support the findings of this study are available on request from the corresponding author [S.M. or F.J.T.].

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
