# Peer review of "Impact of the Gestational Diabetes Diagnostic Criteria during the Pandemic: An Observational Study"

_jcm, 2021, doi:10.3390/jcm10214904_

Round 1
Reviewer 1 Report
Molina-Vega et al show interesting results on GDM diagnosis using alternative criteria during the pandemic. The merit of the study is its prostective nature, as the pandemic forced clinics to alter their diagnostics.
I found it hard to put the current research in context with the previous ones. This was mainly due to the current population consisting of those with failed primary screen for GDM, so therefore all women in this study were at high pretest probability of GDM and was a selected population. In the discussion the authors referred other studies using alternate criteria but did not specify if these studies also were of prescreened women with one failed OGTT or of general pregnant population. This information helps put the current study in perspective. The authors should also discuss their selection bias in more detail.
Despite this these results are interesting and suggest that one can more easily diagnose GDM among those already failing a screening test.
More minor comments:
Sentence on line 45 onwards is hard to understand, please reword.
For HbA1c one should also report mmol/mol.
In the results can you provide the numbers for glucose in mmol/l as well?
Please clarify the p-values in table 1, as it was not clear by table itself that the values were comparing the groups within a year.
Discussion would be easier to read without abbreviations RPG and FPG - maybe say random PG and fasting PG
Reviewer 2 Report
This interesting study assessed the impact of implementing new criteria for diagnosing GDM during the COVID-19 pandemic. In particular, they assessed the role of HbA1c, in comparison to the OGTT. This is a regular practice during this pandemic in many hospitals of the UK, but without evidence on the efficacy of the new strategy. Hence, this study has merit in the current literature. However, some revisions are needed:
- Introduction: I would like to see a paragraph summarizing what the guidelines recommend on the diagnostic criteria for gestational diabetes. Please cite a recent review of guidelines and comment on: PMID: 34192341
- Why did you use 100mg/dl in the first and 95mg/dl in the second trimester of pregnancy, as diagnostic of GDM? Where did you find these cutoffs? Same for HbA1c.
- I think that you need ethical approval, since a new hospital strategy, that could affect patients’ health, was implemented.
- How did you estimate the sample size?
- How did you assess the risk of bias?
- Conclusions: When you try to generalize your results please do it with caution, because we are not sure if it works in the general pregnant population of other areas.
Round 2
Reviewer 2 Report
Thank you for the prompt revisions. I recommend acceptance of the article in its present form.